# Cohort study protocol of the Brazilian collaborative research network on COVID-19: strengthening WHO global data

Fernando Anschau ,[1,2] Natália Del' Angelo Aredes,[3,4] Ludovic Reveiz,[5] Monica Padilla,[6] Rosane de Mendonça Gomes,[6] Wellington Mendes Carvalho,[6] Fernando Antonio Gomes Leles ,[6] Fernanda Baeumle Reese,[7] André Hostílio Hubert,[7] Elisandréa Sguario Kemper,[8] Renilson Rehem de Souza,[8] Cristiane Feitosa Salviano ,[8] Hevelin Silveira e Silva,[8] Eduardo Barbosa Coelho,[9] Giuseppe Cesare Gatto,[4] Rafael Freitas de Morais,[4] Leonardo Nunes Alegre,[10] Rodrigo Citton Padilha dos Reis,[11,12] Joaquim Francisco dos Santos Neto,[11,12] Andresa Fontoura Garbini,[13] César Perdomo Purper,[14] Veridiana Baldon dos Santos,[13] Rafaela dos Santos Charão de Almeida,[13] Bruna Donida,[14] Rogério Farias Bitencourt,[14] Luciane Kopittke,[14] Fernanda Costa dos Santos,[13] Raquel Lutkmeier,[13] Daniela dos Reis Carazai,[13] Virgínia Angélica Silveira Reis,[15] Flávio Clemente Deulefeu,[15] Fernanda Gadelha Severino,[15] José Gustavo da Costa Neto,[15] Nirvania do Vale Carvalho,[16] André Jamson Rocha de Andrade,[16] Adriana Melo Teixeira,[17] Olavo Braga Neto,[17] Gabriel Cardozo Muller,[11,12] Ricardo de Souza Kuchenbecker[11,12]

For numbered affiliations see end of article.

**Correspondence to**
Dr Fernando Anschau;
afernando@ghc.com.br

## ABSTRACT

**Introduction** With the COVID-19 pandemic, hospitals in low-income countries were faced with a triple challenge. First, a large number of patients required hospitalisation because of the infection's more severe symptoms. Second, there was a lack of systematic and broad testing policies for early identification of cases. Third, there were weaknesses in the integration of information systems, which led to the need to search for available information from the hospital information systems. Accordingly, it is also important to state that relevant aspects of COVID-19's natural history had not yet been fully clarified. The aim of this research protocol is to present the strategies of a Brazilian network of hospitals to perform systematised data collection on COVID-19 through the WHO platform.

**Methods and analysis** This is a multicentre project among Brazilian hospitals to provide data on COVID-19 through the WHO global platform, which integrates patient care information from different countries. From October 2020 to March 2021, a committee worked on defining a flowchart for this platform, specifying the variables of interest, data extraction standardisation and analysis.

**Ethics and dissemination** This protocol was approved by the Research Ethics Committee (CEP) of the Research Coordinating Center of Brazil (CEP of the Hospital Nossa Senhora da Conceicao), on 29 January 2021, under approval No. 4.515.519 and by the National Research Ethics Commission (CONEP), on 5 February 2021, under approval No. 4.526.456. The project results will be explained in WHO reports and published in international peer-reviewed journals, and summaries will be provided to the funders of the study.

## STRENGTHS AND LIMITATIONS OF THIS STUDY

⇒ The information generated by this research allows the development of maps of the evolution of SARS-CoV-2 infection.
⇒ This research collected information, which enables the development of public policies to face pandemics.
⇒ This research generated data covering different moments of the COVID-19 pandemic; it has data obtained in the SARS-COV prevaccination era and after the vaccines.
⇒ This research is an observational study.
⇒ This research involves a convenience and non-probabilistic sample of hospitalised patients, and then it may not represent the population of hospitalised patients with COVID-19.

## INTRODUCTION

COVID-19 is a disease caused by SARS-CoV-2, which started in Wuhan (Hubei province, China) in December 2019,[1] and it was characterised as a pandemic by the WHO on 11 March 2020.[2] The disease caused by

SARS-CoV-2 has a lower lethality when compared with that of two other coronaviruses that have already caused epidemics, namely SARS-CoV and MERS-CoV. However, its high transmission rate places a great demand on health systems, with an increase in the number of hospitalisations, need for oxygen by assisted ventilation and other forms of clinical support.[3–5]

Brazil ranks third worldwide in the total number of COVID-19 cases, with all regions affected,[6 7] and demographic and social inequalities impacted the results obtained in transmission control and healthcare. Low-income and middle-income countries, such as Brazil, had great difficulty adopting pandemic mitigating measures such as non-pharmacological interventions.[8] Along with the high transmissibility of variants of interest, so-called *variants of concern*, the pandemic's increased extent was understandable.[9 10]

Brazil is a country of continental dimensions, with 6820 hospitals, of which 2423 are public hospitals and 4397 are private hospitals.[11] The country has 536 474 hospital beds (1.96 and 0.8 beds per 1000 inhabitants in private and public hospitals, respectively).[11 12] There are disparities in the availability of hospital beds in the different regions, including those in the intensive care unit (ICU), which is of utmost importance for the care of patients with COVID-19. For every 100 000 inhabitants, there are 9.73 ICU beds in the Unified Health System (Sistema Único de Saúde (SUS)) in the North region, 10.46 in the Northeast region, 11.87 in the Central-West region, 13.24 in the Southeast region and 15.88 in the South region, according to data from 2020.[13]

Between December 2019 and April 2020, ICU beds were expanded from 46 045 to 60 265[13] nationwide with an overall increase of 30.88%, with the increase in ICU beds under SUS being almost four times smaller than the increase in ICU beds in the private network (13.46% increase under SUS and 48.33% in the private sector), signalling a need for public investment, especially in view of the gaps identified during the pandemic.

A study that evaluated the first 250 000 SARS-Cov-2 hospitalisations in Brazil from the national Severe Acute Respiratory Syndrome case database (SIVEP-Gripe) highlighted the in-hospital mortality rate, which reached 38%, the number of people with demands compatible with the ICU (39%) and regional discrepancies, revealing considerable challenges in managing the pandemic in Brazil.[14] However, the results showed a high proportion of missing data for some variables, especially comorbidities and symptoms, and non-standardised reporting protocols.[14] Additionally, these data did not include the so-called 'second wave', caused predominantly in Brazil by new highly transmissible variants such as P1 and B1.1.7.[15–17]

Data from the two waves observed in Brazil using the same dataset (SIVEP-Gripe) were published in 2021. The cross-sectional study of hospitalised patients with RT-PCR positive for SARS-CoV-2 in Brazil comprised data entered into SIVEP-Gripe from 25 February 2020 to 30 April 2021, separated into two waves on 5 November 2020. As opposed to what was seen in the rest of the world, in-hospital mortality in Brazil increased from 34.81% in the first wave to 39.30% in the second wave. In the second wave, there were more ICU admissions, use of non-invasive and invasive ventilation and increased mortality in the younger age groups. The Brazilian South and Southeast regions had the highest hospitalisation rates per 100 000 inhabitants. Nevertheless, the in-hospital mortality rate was higher in the northern and northeastern states. Racial differences were observed in clinical outcomes, with white people being the most hospitalised, though black/brown-skinned people had higher mortality rates. The younger age groups displayed more considerable differences in mortality compared with the groups with and without comorbidities in both waves. The results reflect the inequalities observed in Brazil and the overload of health systems on the second wave's arrival.[18]

It is critical to maintain surveillance on the pandemic's evolution over time and across the country to measure its impact on the healthcare system. In Brazil, SUS consists of a public network of services at all levels of care, on which about 75% of the population depends exclusively. Today, 66.34% of hospital beds belong to SUS,[19] which characterises the Brazilian public system as responsible for most of the healthcare during the COVID-19 pandemic in the country. Even so, both public and private hospitals do not use a unified electronic record system, which increases the heterogeneity of reports and represents an obstacle for data collection in research.

As a strategy to overcome the challenges of data collection for health research and management purposes, the Pan American Health Organization (PAHO)-WHO proposed a collaboration in 2020 between hospitals in all regions of Brazil to join databases and collect clinical data from patients with COVID-19. Through this initiative, it was possible to gather data from different locations around the country, and the world, to characterise the disease's natural history, identify risk factors for severe cases and describe treatment interventions and outcomes. Thus, it directly contributes to national and global public health with scientific evidence on COVID-19 clinical management and the strengthening of research networks and technology-supported data collection.

As far as we know, this is the first initiative in Brazil to create a 'Big Data' database in healthcare from hospitals and, afterwards, a *data hub* with clinical conditions and usage patterns from the healthcare system.

The aim of this study is to provide standardised quality clinical data on the COVID-19 pandemic in Brazil. Furthermore, it aims to describe the resources and methods used in this collaboration to identify and validate the cases in each database, with patients confirmed as infected with SARS-CoV-2 by laboratory tests (case group) and suspects (control group).

**Table 1** Project research questions and hypotheses

| Questions | Hypotheses |
|---|---|
| Are the descriptions presented in the international literature regarding infections by the SARS-COV-2 applicable to the Brazilian reality? | The descriptions presented in the existing academic literature on the clinical, laboratory, radiological and therapeutic profile of patients diagnosed with the SARS-COV-2 and admitted to Brazilian hospitals are different, a factor that makes it essential to consider Brazilian peculiarities during this disease's screening, diagnosis and treatment. |
| What are the main symptoms presented by Brazilian patients with COVID-19, especially those admitted to different hospital units in Brazil? | The most common clinical manifestations in Brazilian patients tested for SARS-COV-2 are related to fever, myalgia, respiratory symptoms and headache. |
| Is in-hospital mortality from COVID-19 similar in Brazil and the rest of the world? | The in-hospital mortality profile of patients admitted to Brazilian hospitals with COVID-19 is not similar to that presented in the literature. |

## METHODS

This study is being conducted in 43 hospitals located in all regions of Brazil. From March 2020 to June 2022, a committee of experts worked on defining a blueprint for this platform, specifying variables of interest, data extraction standardisation and data analysis.

Aiming to standardise data collection, the project uses *case report forms* (CRF) to extract clinical information from adults and children hospitalised with COVID-19 (accessed from the platform URL: https://www.who.int/teams/health-care-prontidão-clínica-unidade/COVID-19/data platform). The data from all the centres surveyed can be gathered and analysed—either together or by centre or region of the country—to provide a better understanding of the disease and determine the public health response to the pandemic. Collecting anonymised clinical data globally will provide WHO the following:

1. Identification of the main clinical characteristics and prognostic factors of hospitalisation cases for suspected or confirmed COVID-19, expanding our knowledge of the severity, spectrum and impact of the disease on the hospitalised population globally and in different countries.
2. Identification of clinical interventions, facilitating global and country operational planning during the COVID-19 pandemic.

The research questions and hypotheses of the Brazilian protocol of the global clinical platform on COVID-19 are presented in table 1.

The primary and secondary objectives of the Brazilian protocol of the global clinical Platform on COVID-19 are as follows:

► Primary objective: to determine the clinical, laboratory and radiological profile, therapy and mortality of patients confirmed having COVID-19 infection and admitted to Brazilian hospitals.
► Secondary objectives: (1) to identify outcomes such as ICU admission, length of hospital stay and duration of mechanical ventilation and of respiratory therapy; (2) to assess the incidence of adult respiratory dysfunction syndrome and need for mechanical ventilation and intensive care, and days of intensive care in patients admitted to hospitals and (3) to identify the incidence of signs and symptoms of multisystemic inflammatory syndrome and associated chronic diseases.

### Study design, population and sample

This is a multicentre cohort study with convenience sampling from the population of patients hospitalised with COVID-19, because of either clinical suspicion or laboratory confirmation. As inclusion criteria for cases, patients with confirmed COVID-19 (RT-PCR tests or rapid tests), admitted to hospitals with a minimum length of stay of 24 hours, are considered. Patients admitted to one of the hospital units who tested positive for COVID-19 in the period from March 2020 to June 2022 will be entered in the study retrospectively.

### Main outcomes and parameters

The CRFs were developed by WHO for the collection and submission of harmonised data to the COVID-19 global clinical platform, and the procedures for further data collection were performed by the committee's technical Team.

Key descriptive parameters include sociodemographic data (age and sex), presence of pre-existing conditions, use of chronic medications, clinical characteristics on admission and during hospitalisation, laboratory findings on admission and during hospitalisation, clinical interventions on admission and during hospitalisation (use of oxygen, use of mechanical ventilation, use of therapeutics) and patient outcomes (death, discharge and referral).

The patient outcomes described above will be used in secondary analysis to investigate associations between baseline characteristics and disease severity and outcomes.

### Statistical considerations

#### General considerations

Statistical analyses (numerical summaries, point estimates, confidence intervals, hypothesis tests, fitting regression models, tables, graphs and maps) will be performed in R[3] V.4.0.5 or higher. Specification of sets of adjustment variables in certain association analyses will be conducted in DAGitty[4] software.

## Missing data

For each analysis, the denominator will represent the number of available observations.

The date of hospital admission will be considered essential and critical to know the hospitalised population to be analysed. Records with missing admission dates will be excluded from the analysis unless there is confirmation from participating hospitals that the data refer to a cohort of hospitalised patients.

Data imputation will not be performed but can be considered for each particular analysis.

## Descriptive statistics

According to the type of variable, the descriptive statistics will be as follows:

► Quantitative variables: number of observations (N), mean, SD, median and IQR, depending on the distribution of the variable.
► Qualitative variables: number of observations (N), absolute frequency (n), and relative frequency (%).
► Percentages will be calculated for the number of participants with observed data (n/N).

## Objectives of the analysis

See online supplemental table 1.

### Description of sociodemographic and clinical characteristics

To describe the sociodemographic and clinical characteristics of hospital admissions, interventions and clinical outcomes among:

► people infected with the SARS-CoV-2 virus cared for by hospitals within the global platform;
► specific population groups, such as children, pregnant women and populations with co-infections or comorbidities, among others.

### Characterisation of access profiles and hospital resource utilisation patterns

To describe access profiles and hospital resource utilisation patterns such as emergency care, inpatient admissions in wards and ICU, including non-invasive and invasive ventilation devices and non-traditional care modalities such as intermediate units, ventilatory support and others.

### Association between clinical/sociodemographic characteristics and outcomes

To investigate factors associated with disease severity, ICU admission and in-hospital mortality among the following:

► people infected with the SARS-CoV-2 virus cared for by hospitals within the global Platform;
► specific population groups, as described above.

### Sources of inequities

To investigate potential sources of inequities in timely access to diagnostic and therapeutic services and resources and the association with clinical outcomes.

## Data governance

Defining data governance is always necessary in a research protocol, considering that data are protected by legislation, including the rights of research participants. This model dictates that the hospitals involved understand the research aspects, the information security points and complete the process of sending data to the central repository in a clear, unambiguous manner. Furthermore, regarding information security, the central repository's teams are required to follow procedures ensuring ethical and secure actions, in compliance with the legislation in force, when processing information.

Privacy and consent policies are important points for discussion on health data sharing. Therefore, for the preparation of this proposal, the concepts and technical principles of the General Data Protection Act (LGPD) were respected, considering health data as sensitive, as described in these documents.[20] The LGPD regulates the processing of personal data and aims to protect citizens' right to freedom, privacy and free development.

Considering all these aspects, three levels of governance/access were defined: strategic, tactical and operational:

1. The Advisory Committee of the PAHO/WHO COVID-19 Platform Project represents the first, whose main objective is to technically support PAHO/WHO in the terms of the COVID-19 global clinical data platform, and the participating institutions in the activities of preparation for analysis and publication of clinical and epidemiological data.
2. The second consists of those responsible for translating the strategy and integrating all parties, setting the objectives and managing the project, including defining the work processes, schedule, team engagement and communication, quality and results follow-up;
3. The third performs activities from the tactical level always in compliance with data security standards and rules. It is at this level that statements like 'data must be stored securely' translate into technological tools ensuring the semantics read into the sentence.

## Risks

Risks are related to the loss of confidentiality. However, they have been and will be constantly mitigated/controlled by the anonymisation of the participants' identification data. The researchers take responsibility for maintaining their commitment to preserve data security and confidentiality, and hospital institutions are asked to take all necessary steps to protect their access credentials and passwords to the COVID-19 data platform.

The collected and anonymised data will be stored by the aforementioned WHO platform, a secure, password-protected, restricted-access electronic platform. WHO commits to the following: (i) to protect the confidentiality of anonymous data on COVID-19 and to prevent its unauthorised disclosure; (ii) to implement and maintain technical and organisational security measures to protect the security of anonymous data on

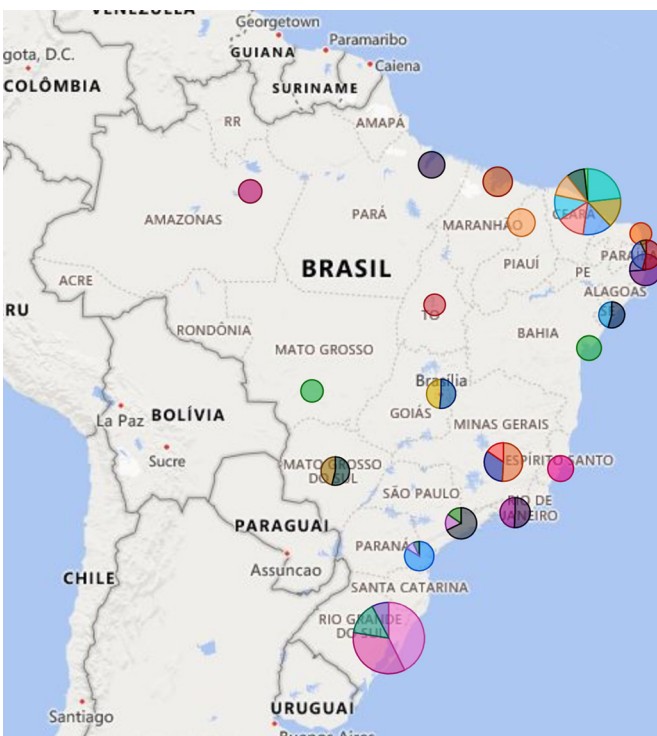

**Figure 1** Hospitals participating in the survey by size classification.

the platform and (iii) to follow the International Health Regulations.[21]

## Benefits

It is expected that this protocol will produce specific and reliable knowledge of the COVID-19 pandemic—clinical and epidemiological analysis, leading to discussions on

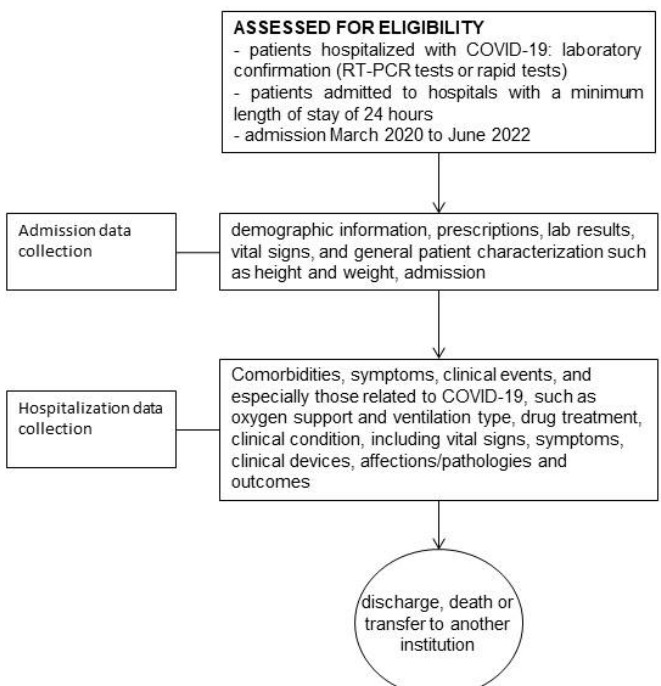

**Figure 2** Illustration of flow diagram of study protocol.

Brazil and other countries and their reality and plans to fight the pandemic. It is expected that it will deepen our knowledge of the pandemic scenario and help hospital institutions to develop preventive measures and health service protocols and to strengthen staff training in existing complications. The information generated by the research can serve as a basis for developing maps of the SARS-CoV-2 infection evolution.

## Clinical data extraction

As the WHO representative in the country, the PAHO/WHO office in Brazil, through the Technical Unit of Health Systems and Services, coordinated the project by dialoguing with the partner institutions to make this project effective. In this context, Grupo Hospitalar Conceição became the national coordinating Brazilian institution for submission procedures to the Research Ethics Committee (CEP) and the National Research Ethics Commission (CONEP).

Data collection started retrospectively in 43 Brazilian hospitals—as shown in figure 1—starting from the first COVID-19 case diagnosed in Brazil (March 2020) and ending in June 2022. We chose to carry out this first stage of data collection directly in the hospital units, consulting the databases of the clinical and care information systems available in each institution or the physical records of the hospitalisations of patients with suspected or confirmed COVID-19 in these units. Figure 2 shows the flowchart of the study protocol.

The selected hospitals are responsible for providing all the conditions for the project's execution, such as access to data collection and feeding the platform to the PAHO/WHO consultant assigned to each institution. Accordingly, each hospital signed a commitment form that contains all the guarantees offered by the WHO to maintain data confidentiality and anonymisation. The collection and access process, in agreement with each hospital unit, respects the specificities and internal regulations and also follows the ethical and data security norms.

Considering that the information contained in the hospital records of each institution can be divided between structured and open text, and in view of the heterogeneities of Brazilian hospitals in terms of management, there is a need to organise data collection in several ways: one through manual consultation of electronic citizen records (ECR)—or physical records for hospitals that do not have ECRs—and filling out the WHO-specific forms and another through the development of an automated programming interface (API) for filling out these forms.

The standardised API was developed by information technology (IT) professionals to share databases from different hospitals. These databases, with provision for secure cloud-based storage, contain deidentified electronic health records corresponding to admission notes for the first 48 hours of care and the 24 hours prior to discharge/outcome for each research participant.

The structured ECR data contain demographic information, prescriptions, laboratory results, vital signs and

general patient characterisation such as height and weight and admission and outcome dates. Since the lack of data for different variables can be significant, there is the possibility for institutions to search and fill in part of the variables from unstructured data by open text extraction.

For this procedure, the Smart Health Connect (SHC) software is used, which operates on the basis of an algorithm using deep neural networks, extracting information from the clinical evolutions and transferring to the CRF data that were previously missing, reinforcing the completeness of the database exported to the WHO. The main variables of interest incorporated through this software were comorbidities, symptoms and clinical events, especially those related to COVID-19, such as oxygen support and ventilation type, and outcomes. The software is capable of recognising and categorising data regarding history (past), current events (present) and expected actions or outcomes (future) and identifying variables related to drug treatment and clinical condition, including vital signs, symptoms, clinical devices, affections/pathologies and other relevant data that could not be captured in the structured ECR data fields.

SHC uses the newest natural language processing techniques available to the scientific community and a software architecture model that enables mass data extraction.

Regarding the technological standard used by the software, there are two approaches supported by artificial intelligence. The first is named entity recognition, which has the function of locating previously defined entities in open text. For this project, the categories are disease names, signs and symptoms, procedures, medications, ventilatory support and test results. SHC identifies and marks in the text each of the identified categories in a process known as inference.

Once the relevant terms are identified, the other step is the analysis of the findings in a given text, called assertion detection. Through this approach, it is possible to differentiate the status of what has or has not actually occurred with that patient. This is necessary because the medical records can indicate something related to the past, or to a third party, such as family history, or even state that the patient does not have a certain condition or does not use a certain medication.

## Expected results

At the end of the project, in addition to the reports produced by WHO with the data provided globally and the ensured access to the databases, PAHO/WHO in Brazil will organise publications in partnership with the institutions participating in the project in the country, highlighting the results and Brazil's contribution in fighting the COVID-19 pandemic.

Based on this protocol design, it will be possible to continue data collection among the participating hospitals and continuously add valuable information for public health in the context of facing the pandemic in Brazil. Once collaboration between institutions has taken place and the form of data collection, variables of interest

and technologies to support the search for information of interest have been standardised, there is significant potential to advance the analysis of evidence and response measures in the national health system.

Until now, the identification of this strategy's partner hospitals and their characterisation, with number and type of beds, provide an analysis perspective of the healthcare network scenario in the fight against the pandemic (table 2).

## Ethics

This protocol was approved by the Research Ethics Committee (CEP) of the Research Coordinating Center of Brazil (CEP of the Hospital Nossa Senhora da Conceição), on 29 January 2021, under approval No. 4.515.519 and by the National Research Ethics Commission (CONEP), on 5 February 2021, under approval No. 4.526.456. Afterwards, it received the approval of the local CEPs of each institution involved. In addition, this study follows the resolutions 466/12 and 510/16 and the resolutions CNS No. 466/2012 and No. 510/2016 of the National Health Council, which deal with the ethical aspects involving the use of data directly obtained from participants or identifiable information, or that may entail risks greater than those existing in everyday life. The results of this study will be published in international peer-reviewed journals and summaries will be provided to the funders of the study.

## Strengths and limitations

Through this initiative, it is possible to gather data from different locations around the country to characterise the disease's natural history, identify risk factors for severe cases and describe treatment interventions and outcomes. Thus, it directly contributes to national and global public health with scientific evidence on COVID-19 clinical management and to the strengthening of research networks and technology-supported data collection.

As far as we know, this is the first initiative in Brazil to create a 'Big Data' database in healthcare from hospitals and, afterwards, a *data hub* with clinical conditions and usage patterns from the healthcare system.

Because this is a convenience, non-probabilistic sample of patients admitted to healthcare facilities, it may not represent the population of patients with COVID-19 hospitalised in all the country. However, the information generated by this research can serve as a basis for the development of maps of the evolution of SARS-CoV-2 infection and public policies to face pandemics.

## Patient and public involvement

This project is already involving both Brazilian communities and public agencies that are facing the fight against the COVID-19 pandemic as well as the Brazilian Ministry of Health and PAHO/WHO. We intend to expand partnerships and involve more stakeholders from Brazil.

 Anschau F, *et al. BMJ Open* 2022;**12**:e062169. doi:10.1136/bmjopen-2022-062169

**Table 2** WHO global clinical platform for COVID-19 characterisation in Brazil: collaborating centres

| State | Hospital institution | Hospital beds | | | | | | Hospital day | Total |
|---|---|---|---|---|---|---|---|---|---|
| | | Surgical | Clinical | Complementary | Obstetrical | Paediatric | Medical specialties | | |
| AM | HU Getúlio Vargas (HUGV-UFAM) | 58 | 59 | 32 | 0 | 0 | 0 | 0 | 149 |
| PA | Hospital Universitário João de Barros Barreto (HUJBB-UFPA) | 40 | 178 | 38 | 0 | 28 | 12 | 10 | 306 |
| CE | HU Walter Cantídio (HUWC-UFC) | 68 | 72 | 21 | 0 | 20 | 22 | 4 | 207 |
| CE | Maternidade Escola Assis Chateaubriand (MEAC-UFC) | 21 | 58 | 65 | 104 | 6 | 0 | 0 | 254 |
| CE | Hospital Regional Norte (HRN-ISGH) | 58 | 183 | 177 | 33 | 70 | 16 | 0 | 537 |
| CE | Hospital Geral Waldemar Alcântara (HGWA-ISGH) | 18 | 188 | 58 | 0 | 33 | 33 | 0 | 330 |
| CE | Hospital Regional do Cariri (HRC-ISGH) | 88 | 112 | 113 | 0 | 0 | 29 | 0 | 342 |
| CE | Hospital Regional do Sertão Central (HRSC-ISGH) | 58 | 104 | 106 | 30 | 1 | 14 | 0 | 313 |
| CE | Hospital Estadual Leonardo Da Vinci (Helv-ISGH) | 1 | 112 | 179 | 0 | 0 | 0 | 0 | 292 |
| CE | Hospital Regional Vale do Jaguaribe (HRVJ-ISGH) | 0 | 30 | 10 | 0 | 0 | 0 | 0 | 40 |
| DF | HU da Universidade de Brasília (HUB-UnB) | 49 | 76 | 41 | 30 | 20 | 3 | 12 | 231 |
| ES | HU Cassiano Antônio Moraes (HUCAM-UFES) | 82 | 90 | 55 | 20 | 24 | 0 | 2 | 273 |
| MA | HU da Universidade Federal do Maranhão | 134 | 75 | 81 | 71 | 92 | 0 | 0 | 453 |
| MT | HU Julio Muller (HUJM-UFMT) | 12 | 20 | 31 | 24 | 11 | 0 | 0 | 98 |
| MS | HU Maria Aparecida Pedrossian (Humap-UFMS) | 60 | 45 | 53 | 32 | 18 | 0 | 12 | 220 |
| MS | HU da Univ. Federal da Grande Dourados (HU-UFGD) | 29 | 41 | 59 | 25 | 35 | 1 | 0 | 190 |
| MG | HC da Univ. Federal do Triangulo Mineiro (HC-UFTM) | 93 | 86 | 57 | 22 | 42 | 2 | 8 | 310 |
| MG | HU da Univ. Federal de Juiz de Fora (HU-UFJF) | 19 | 61 | 29 | 0 | 17 | 0 | 17 | 143 |
| MG | HC da Univ. Federal de Minas Gerais (HC-UFMG) | 119 | 114 | 107 | 35 | 63 | 1 | 30 | 469 |
| PB | HU Lauro Wanderley (HULW-UFPB) | 31 | 80 | 43 | 36 | 22 | 0 | 33 | 245 |
| PE | HU da Univ. Fed. Vale do S. Francisco (HU-UNIVASF) | 86 | 35 | 21 | 0 | 0 | 8 | 0 | 150 |
| PE | HC da Univ. Federal de Pernambuco (HC-UFPE) | 118 | 119 | 27 | 30 | 38 | 13 | 76 | 421 |
| RJ | HU Antônio Pedro (HUAP-UFF) | 83 | 54 | 59 | 21 | 17 | 0 | 10 | 244 |
| RN | HU Ana Bezerra (HUAB-UFRN) | 8 | 6 | 10 | 33 | 10 | 0 | 0 | 67 |
| RS | HE da Univers. Feder. de Pelotas (HE-UFPel) | 8 | 109 | 31 | 21 | 20 | 0 | 10 | 199 |
| RS | HU da Univ. Federal de Santa Maria (HUSM-UFSM) | 75 | 163 | 60 | 35 | 41 | 1 | 10 | 385 |
| SP | HU da Univ. Federal de São Carlos (HU-UFSCAR) | 0 | 56 | 12 | 0 | 17 | 0 | 0 | 85 |

Continued

**Table 2** Continued

| State | Hospital institution | Hospital beds | | | | | | | | Total |
|---|---|---|---|---|---|---|---|---|---|---|
| | | Surgical | Clinical | Complementary | Obstetrical | Paediatric | Medical specialties | Hospital day | |
| PB | HU Alcides Carneiro (HUAC-UFCG) | 29 | 84 | 31 | 0 | 33 | 0 | 0 | 177 |
| PB | HU Julio Bandeira (HUJB-UFCG) | 12 | 2 | 1 | 0 | 15 | 0 | 2 | 32 |
| RJ | HU Gaffree e Guinle (HUGG-UNIRIO) | 95 | 89 | 38 | 19 | 6 | 2 | 3 | 252 |
| SE | HU da Universidade Federal de Sergipe (HU-UFS) | 49 | 48 | 25 | 0 | 14 | 10 | 10 | 156 |
| SE | Hospital Universitário de Lagarto (HUL-UFS) | 25 | 46 | 33 | 0 | 16 | 0 | 10 | 130 |
| TO | Hospital de Doenças Tropicais (HDT-UFT) | 0 | 37 | 2 | 0 | 9 | 0 | 6 | 54 |
| BA | Hospital Especializado Octávio Mangabeira | 23 | 105 | 28 | 0 | 17 | 42 | 1 | 216 |
| PR | Complexo Hospitalar do Trabalhador | 126 | 37 | 132 | 35 | 30 | 24 | 0 | 384 |
| PR | Hospital do Trabalhador | 4 | 22 | 0 | 2 | 4 | 0 | 0 | 32 |
| PR | Hospital Osvaldo Cruz | 0 | 29 | 0 | 0 | 1 | 4 | 8 | 42 |
| DF | Hospital da Criança de Brasília José Alencar | 0 | 0 | 58 | 0 | 150 | 2 | 8 | 218 |
| PI | Hospital Getúlio Vargas | 170 | 87 | 94 | 0 | 0 | 2 | 11 | 364 |
| RS | Hospital Nossa Senhora da Conceição—Grupo Hospitalar Conceição | 207 | 357 | 332 | 62 | 105 | 30 | 13 | 1106 |
| RS | Hospital de Clínicas de Porto Alegre—HCPA/ UFRGS | 228 | 224 | 259 | 50 | 95 | 30 | 17 | 903 |
| SP | Instituto de Infectologia Emilio Ribas, São Paulo | 4 | 82 | 232 | 0 | 16 | 10 | 31 | 375 |
| SP | Hospital Nestor Goulart Reis Americo Brasiliense | 0 | 20 | 0 | 0 | 0 | 73 | 0 | 93 |
| Total | | 2388 | 3595 | 2840 | 770 | 1156 | 384 | 354 | 11 487 |

Source: The Brazilian National Register of Health Service Providers (CNES).[13]

AM, Amazonas; BA, Bahia; CE, Ceará; DF, Distrito Federal; ES, Espírito Santo; HE, Teaching Hospital; HU, University Hospital; MA, Maranhão; MG, Minas Gerais; MS, Mato Grosso do Sul; MT, Mato Grosso; PA, Pará; PA, Pará; PB, Paraíba; PE, Pernambuco; PI, Piauí; RJ, Rio de Janeiro; RN, Rio Grande do Norte; RS, Rio Grande do Sul; SE, Sergipe; SP, São Paulo; TO, Tocantins.

## DISCUSSION

One of the main concerns regarding the COVID-19 pandemic is its burden on the organisation of the healthcare system.[22] The need for a general and effective organisation of health services to treat the population safely and fight the disease's complications has required a great deal of effort. This includes reallocating healthcare workers, using appropriate medicines, producing, distributing and delivering vaccines to the population and increasing healthcare capacity.[23]

The outbreak of the highly transmissible delta variant (B.1.617.2) raises new questions among policy makers and health system managers regarding the impact on hospitalisation rates, a point reinforced by a recent published study that indicates a possible worst-case scenario compared with the alpha variant (B.1.1.7).[24] Additionally, according to Twohig et al,[24] this recent research adds to previous evidence and reinforces the notion that the delta variant, in unvaccinated populations, may increase the burden on the overall healthcare system when compared with the SARS-CoV-2 alpha variant.[24]

The Brazilian ICU occupancy condition became critical in most of the country in March 2021, when 20 states reached critical rates (equal to or greater than 80% occupancy) and became intermediate in most states by June 2021 (between 60 and 80%), in concurrence with high SARS-CoV-2 transmission rates.[25 26]

It is necessary to maintain biosafety education for the general population (social distancing and the use of masks), encourage them to seek the vaccine when called on by age/group and complete the recommended number of doses. At the organisational level of hospitals, it is critical to measure resources and indicators, planning strategies according to the literature and lessons learnt during the pandemic to meet the next demand. For the public policy level, it is crucial to intensify vaccination throughout the country and support the health system by consolidating its structure and strategies for serving the population.

The initial ignorance regarding the virus and the associated disease led hospitals and health services to seek different internal management strategies to meet the demand coming with COVID-19.[27] Berwick used the term 'choices for the new normal' to point to six properties of care for lasting change: time, standards, working conditions, proximity, readiness and equity. For the author, the changes in care required by the pandemic may allow health systems to become more patient-centred and efficient and consequently provide better care to the public.[28]

Since the spectrum of SARS-CoV-2 infection can range from asymptomatic infection to potentially fatal complications of COVID-19, we chose to list patients with suspected and confirmed infection who were admitted to hospital units.[29] Thus, we believe that the strategy of the 'COVID-19 Global Clinical Data Platform for clinical characterisation and management of hospitalized patients with suspected or confirmed COVID-19' contributes to the understanding of the Brazilian context of pandemic crisis in the healthcare system by providing data analysis of clinical aspects and epidemiology.

It also reinforces a portrait of the reality of the institutions based on the current demand, discussion on aspects of the main organisational frameworks for fighting the pandemic and how to keep the institutions functioning and meeting the health needs of the population amidst the challenges.

## CONCLUSION

This project describes the collaboration of several Brazilian hospitals to build a joint database, with special attention to protocols aimed at reducing heterogeneity between different platforms and ensuring the quality of research data.

As part of the process, the project management carried out an organised multicentre research framework involving PAHO/WHO, as the centre of the strategy, and Brazilian public hospitals, resulting in substantial data collection. In addition, the data extraction used innovative techniques in the technology area, promoting quality and mitigating the lack of relevant information, moving towards data collection in unstructured fields, such as clinical evolution which are the texts with the most detailed health data in the medical records.

This proposal accomplishes an important step towards fighting COVID-19 in a developing country of continental dimensions such as Brazil, by building a foundation for new collaborations between referral hospitals and demonstrating the feasibility of joint initiatives for the emergence of 'Big Data' analysis strategies.

**Author affiliations**
[1]Research Sector, Grupo Hospitalar Conceição, Porto Alegre, Brazil
[2]Programa de Pós Graduação em Neurociências—ICBS, Universidade Federal do Rio Grande do Sul, Porto Alegre, Brazil
[3]Nursing School, Universidade Federal de Goiás, Goiania, Brazil
[4]Brazilian Company of Hospital Services, Brasilia, Brazil
[5]Pan American Health Organization (PAHO)—World Health Organization (WHO), Washington, District of Columbia, USA
[6]Health Systems and Services (HSS) - Pan American Health Organization (PAHO) - World Health Organization (WHO), Brasilia, Brazil
[7]Complexo Hospitalar do Trabalhador, Curitiba, Brazil
[8]Hospital da Criança de Brasília, Brasilia, Brazil
[9]Brazilian Company of Hospital Services, Belo Horizonte, Brazil
[10]Brazilian Company of Hospital Services, Porto Alegre, Brazil
[11]Graduate Program in Epidemiology, Universidade Federal do Rio Grande do Sul, Porto Alegre, Brazil
[12]Hospital de Clínicas de Porto Alegre, Porto Alegre, Brazil
[13]Grupo Hospitalar Conceição, Porto Alegre, Brazil
[14]Grupo Hospitalar Conceicao Gerencia de Ensino e Pesquisa, Porto Alegre, Brazil
[15]Instituto de Saúde e Gestão Hospitalar, Fortaleza, Brazil
[16]Hospital Getúlio Vargas, Health Department of the State of Piauí, Teresina, Brazil
[17]Department of Hospital, Home, and Emergency Care (DAHU) of the Specialized Health Care Office (SAES), Brazilian Ministry of Health, Brasilia, Brazil

**Acknowledgements** This project received the institutional support from the hospital institutions mentioned here through access to data and research participants, which were critical for this research, the reason for our gratitude. The

authors would like to thank the research participants who made the development of the research project possible.

**Contributors** A number of people were involved in the development of the research proposal. FA, BD, RdSK and LK contributed to the organisation of the group of researchers, data collection and analysis and writing of the protocol. NDAA made substantial contributions to the design of the work, data collection based on the protocol (in progress), literature review and writing. She also revised the manuscript and approved the submitted version. LR, MP, RdMG, WMC and FAGL worked on the organisation of the group of researchers and logistical support. FBR, AHH, ESK, RRdS, CFS, HSeS and GCG worked on database organisation and data collection. EBC contributed to writing the paper and data analysis. RFdM, LNA and CPP were also involved with database organisation. GCM contributed to database organisation and statistical analysis. RCPdR, JFdSN, AFG, VBdS, RSCS, FCdS, RL and DdRC contributed with data collection. RFB, AMT and OBN contributed to the organisation of the group of researchers. VASR, FCD, FGS, JGdCN and NVC participated with data collection and database organisation and analysis. Finally, AJRdA contributed with data collection and analysis.

**Funding** The WHO Global Clinical Platform for COVID-19 is supported by WHO (WHO Voluntary Contributions—Emergencies—Grant: 71856-01 Clinical Care).

**Map disclaimer** The inclusion of any map (including the depiction of any boundaries therein), or of any geographic or locational reference, does not imply the expression of any opinion whatsoever on the part of BMJ concerning the legal status of any country, territory, jurisdiction or area or of its authorities. Any such expression remains solely that of the relevant source and is not endorsed by BMJ. Maps are provided without any warranty of any kind, either express or implied.

**Competing interests** None declared.

**Patient and public involvement** Patients and/or the public were involved in the design, or conduct, or reporting, or dissemination plans of this research. Refer to the Methods section for further details.

**Patient consent for publication** Not applicable.

**Provenance and peer review** Not commissioned; externally peer reviewed.

**ORCID iDs**
Fernando Anschau http://orcid.org/0000-0002-2657-5406
Fernando Antonio Gomes Leles http://orcid.org/0000-0002-3891-0443
Cristiane Feitosa Salviano http://orcid.org/0000-0002-0221-6011

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
