## [Reviewer comments · BMJ Open]

ARTICLE DETAILS

TITLE (PROVISIONAL)	Cohort Study Protocol of the Brazilian Collaborative Research Network on COVID-19: strengthening WHO global data
AUTHORS	Anschau, Fernando; Aredes, Natália; Reveiz, Ludovic; Padilla, Monica; Gomes, Rosane; Carvalho, Wellington; Leles, Fernando; Reese, Fernanda; Hubert, André; Kemper, Elisandréa; de Souza, Renilson; Salviano, Cristiane; e Silva, Hevelin; Coelho, Eduardo; Gatto, Giuseppe; de Moraes, Rafael; Alegre, Leonardo; Padilha dos Reis, Rodrigo; dos Santos Neto, Joaquim; Garbini, Andresa; Purper, César; dos Santos, Veridiana; Charão de Almeida, Rafaela; Donida, Bruna; Bitencourt, Rogério; Kopittke, Luciane; dos Santos, Fernanda; Lutkmeier, Raquel; Carazai, Daniela; Reis, Virgínia; Deulefeu, Flávio; Severino, Fernanda; da Costa Neto, José Gustavo; Carvalho, Nirvania; de Andrade, André; Teixeira, Adriana; Braga Neto, Olavo; Muller, Gabriel; Kuchenbecker, Ricardo

VERSION 1 – REVIEW

REVIEWER	Kanagasabai, Udhayashankar Centers for Disease Control and Prevention Center for Global Health, Division of Global HIV and TB
REVIEW RETURNED	19-May-2022

GENERAL COMMENTS	This is a well written protocol that will provide interesting findings that will inform decision making.
--

REVIEWER	Ahmed, Kamran World Health Organization Regional Office for Africa
REVIEW RETURNED	13-Jun-2022

GENERAL COMMENTS	This is a well-written, well-organized and ongoing research protocol, approved by research ethics committee and aimed at presenting the strategies of Brazilian hospital network (43 hospitals) to perform systematized COVID-19 data collection using WHO global platform. This article is an important contribution that expect to deepen hospital/institutional knowledge about the pandemic in Brazil and help in developing preventive measures, health service protocols and strengthen the training of teams in the existing complications. The study claims that this is the first initiative in Brazil to create Big Data database in healthcare system. This contributes to the understanding of the Brazilian context of pandemic crisis in the healthcare system by providing data analysis on clinical aspects and epidemiology. I believe that following comments are for minor revisions but could be strengthened by attention to these areas: Specific comments:
---

	1) The methods section would benefit from illustration of flow diagram of study protocol. 2) In methods (page 8, line #7), it has been mentioned that this study is being conducted in 43 hospitals. On page 15, line #6, it says that first phase of data collection completed on March 2021. Please clarify data collection start and end dates clearly, and if there is a phased approach, include timeline. 3) Please be consistent. Use terms COVID-19, SARS-COV-2 and new coronavirus 2019 consistently. Standard practice is COVID-19 is used for disease and SARS-COV-2 for virus that cause this disease and suggest to use these terms consistently throughout this paper. 4) Please consider following format or please explain if grant number is not available. Funding statement: preferably worded as follows. Either: 'This work was supported by [name of funder] grant number [xxx]' or 'This research received no specific grant from any funding agency in the public, commercial or not-for-profit sectors'.
--	---

VERSION 1 – AUTHOR RESPONSE

Reviewer: 1

Dr. Udhayashankar Kanagasabai, Centers for Disease Control and Prevention Center for Global Health

Comments to the Author:

This is a well written protocol that will provide interesting findings that will inform decision making.

Reviewer: 2

Dr. Kamran Ahmed, Centers for Disease Control and Prevention

Comments to the Author:

This is a well-written, well-organized and ongoing research protocol, approved by research ethics committee and aimed at presenting the strategies of Brazilian hospital network (43 hospitals) to perform systematized COVID-19 data collection using WHO global platform. This article is an important contribution that expect to deepen hospital/institutional knowledge about the pandemic in Brazil and help in developing preventive measures, health service protocols and strengthen the training of teams in the existing complications. The study claims that this is the first initiative in Brazil to create Big Data database in healthcare system. This contributes to the understanding of the Brazilian context of pandemic crisis in the healthcare system by providing data analysis on clinical aspects and epidemiology.

I believe that following comments are for minor revisions but could be strengthened by attention to these areas:

Specific comments:

1) The methods section would benefit from illustration of flow diagram of study protocol. Authors' response: Thanks for the contribution, we made the suggested change.

2) In methods (page 8, line #7), it has been mentioned that this study is being conducted in 43 hospitals. On page 15, line #6, it says that first phase of data collection completed on March 2021. Please clarify data collection start and end dates clearly, and if there is a phased approach, include timeline.

Authors' response: The text has been corrected.

3) Please be consistent. Use terms COVID-19, SARS-COV-2 and new coronavirus 2019 consistently. Standard practice is COVID-19 is used for disease and SARS-COV-2 for virus that cause this disease

and suggest to use these terms consistently throughout this paper. . Authors' response: The change has been made.

4) Please consider following format or please explain if grant number is not available.

Funding statement: preferably worded as follows. Either: 'This work was supported by [name of funder] grant number [xxx]' or 'This research received no specific grant from any funding agency in the public, commercial or not-for-profit sectors'.

Authors' response: Thanks for the contribution, we made the suggested change.

Reviewer: 1

Competing interests of Reviewer: No competing interests

Reviewer: 2

Competing interests of Reviewer: No competing interest